# Clinical ultrasound, photoacoustic, and fluorescence image-guided lymphovenous anastomosis microsurgery via a transparent ultrasound transducer array

Jeongwoo Park [1,2,6], Donghyeon Oh [1,6], Jinhee Yoo [1,6], Honghyeon Ha[1,3], Donggyu Kim[1], Hyung Ham Kim [1] ✉, Yujin Myung[4] ✉ & Chulhong Kim [1,5] ✉

Multimodal optical and ultrasound imaging (USI) provides complementary diagnostic insights. However, because conventional USI uses opaque ultrasound (US) transducers, integrating these two modalities results in a bulky and complicated handheld probe in which neither modality performs efficiently. Although transparent ultrasound transducers (TUTs) solve these issues by acting as optical windows, enabling the seamless combination of light and US beams, single-element TUTs are not common in clinical environments. Here, we demonstrate a clinical triple-modal US, photoacoustic, and fluorescence imaging system, seamlessly integrated via a linear TUT-array. This system, with 64 channels and a 7-MHz center frequency achieves 72.7% optical transparency in the near infrared region. The system's handheld opto-US probe coaxially integrates the TUT-array with a miniaturized camera and an optical fiber in a small form factor. The triple-modal imaging system effectively visualizes tissue structures, vasculatures, and lymphatics in real time in live animals, healthy volunteers, and lymphedema patients. By accurately mapping superficial tissues, blood vessels, and lymphatic vessels, we use the prototype system to successfully guide lymphovenous anastomosis microsurgery. These preclinical demonstrations illustrate the potential use of our system in various clinical procedures requiring microsurgical guidance, paving the way for future advances in multimodal imaging.

In contemporary medical assessments, imaging provides non-invasive insights for early diagnosis, treatment planning, and real-time monitoring of physiological processes[1]. Optical imaging (OI) is particularly valuable in differentiating between tissue types and identifying molecular or cellular activity, allowing us to monitor functional and molecular dynamics[2–4]. Nevertheless, due to the strong light scattering in tissue, the penetration depth of OI is constrained to the subsurface-level[5,6]. On the other hand, ultrasound imaging (USI) can penetrate

[1]Department of Electrical Engineering, Convergence IT Engineering, Mechanical Engineering, Medical Science and Engineering and Medical Device Innovation Center, Pohang University of Science and Technology (POSTECH), Pohang, Republic of Korea. [2]Department of Biomedical Convergence Science and Technology, Advanced Bioconvergence, and Cell and Matrix Research Institute, Kyungpook National University, Daegu, Republic of Korea. [3]SonicLab Inc., Siheung, Republic of Korea. [4]Department of Plastic and Reconstructive Surgery, Seoul National University Bundang Hospital, Seoul National University College of Medicine, Seongnam, Republic of Korea. [5]Opticho Inc., Pohang, Republic of Korea. [6]These authors contributed equally: Jeongwoo Park, Donghyeon Oh, Jinhee Yoo. ✉e-mail: david.kim@postech.ac.kr; surgene@snu.ac.kr; chulhong@postech.edu

deeply into tissues, providing anatomical visualization[7,8]. USI is widely employed in clinical practice for guiding procedures and monitoring dynamic physiological changes, making it indispensable across both primary care and specialized settings[9–12]. However, USI lacks the molecular and functional specificity needed for targeted disease detection and assessment[6,12].

Multimodal imaging combines complementary modalities to overcome individual limitations, synergizing their strengths to deliver comprehensive diagnostic insights. For example, combining positive emission tomography (PET) with computed tomography[13,14] or magnetic resonance imaging (MRI)[15,16] yields both metabolic and anatomical information, overcoming the limited spatial resolution of PET alone while enhancing diagnostic precision through detailed structural imaging. Similarly, the combination of OI and USI seamlessly merges functional and molecular insights with high-resolution, deep-tissue anatomical visualization. Fluorescence (FL) imaging (FLI) with USI offers molecular-targeted *en-face* guidance with anatomical localization, enhancing surgical accuracy[17–22]. Moreover, photoacoustic (PA) imaging (PAI) with USI provides detailed microvascular mapping, oxygen saturation levels, and molecular agent detection along with structural imaging, enabling comprehensive real-time diagnostics[23–27].

Despite the advantages, integrating USI and OI faces several challenges arising from the opacity of conventional ultrasound transducers (UTs). Early solutions involved oblique light paths, which increased the system size and complexity and reduced signal-to-noise ratios (SNRs)[28–32]. An opto-US combiner can address this issue, but it still complicates the system design[33–36]. Another approach uses ring-shaped UTs with central holes for light passage, but this design compromises electrical properties and the image quality[37]. Recently, transparent ultrasound transducers (TUTs) have emerged, allowing light transmission, simplifying system design, and improving SNRs[37–42]. However, most TUTs still rely on single-element designs, requiring mechanical scanning and limiting clinical uses. Although recent developments in TUT-array[43] and transparent capacitive micromachined ultrasound transducers[44] have demonstrated potential, they have yet to be adapted into handheld forms suitable for clinical uses.

In this study, we present a clinical triple-modal USI, PAI, and FLI system, fully integrated in a handheld opto-US probe. This combined multimodal probe incorporates a 64-element TUT-array for USI/PAI, a miniaturized light source for PAI/FLI, and an optical camera for FLI, all in a small handheld device. Designed for a wide range of applications, the TUT-array operates at a 7 MHz center frequency with 45% bandwidth, and its transparency allows for coaxial alignment of light and US beams. It demonstrates feasibility in dual-modal USI/PAI on human arms, mapping vascular structures and hemoglobin oxygen saturation (sO₂) from PAI overlaid on the conventional B-mode US image. In mice, triple-modal USI/PAI/FLI imaging before and after FDA-approved indocyanine green (ICG) injections quantifies signal enhancement in blood and lymphatic vessels. Finally, the triple-modal USI/PAI/FLI demonstrates clinical potential by guiding microsurgical treatment for lymphedema patients, particularly during preoperative planning. In microsurgical lymphaticovenous anastomosis (LVA) procedures, it is essential for clinicians to accurately observe the patient's lymphatic and vascular conditions, as this information is key to planning personalized surgical treatment[45]. Although conventional imaging modalities, such as MRI[46], USI[47], and OI[48,49], have been used for lymphatic assessments, no clinical device has yet been developed for multimodal imaging to allow real-time diagnosis of lymphatics, microvessels, and surrounding tissues. Our triple-modal imaging system provides a comprehensive solution by providing structural information through USI, mapping microvessels and lymphatics with PAI, and visualizing lymphatics *en-face* with FLI, all within a single, seamless system. This triple-modal imaging system not only overcomes the limitations of conventional approaches in reconstructive surgery field but also potentially opens avenues for broader clinical applications, providing

a versatile platform for both surgical planning and non-invasive diagnostics.

## Results

### Fabrication of the TUT-array

As shown in Fig. 1a, the TUT-array is fabricated in five steps: preparing the individual layers, laminating the layers into a stack, curing and stabilizing the stack, slicing the stack into channels, and overcasting the acoustic lens. (1) The first step is preparing the individual components that form the TUT-array. As the piezoelectric material, lead magnesium niobate-lead titanate (PMN-PT) is used for its high electromechanical coupling coefficient (~0.6), high piezoelectric constant (~1190 pC/N), and low longitudinal velocity (~4600 m/S), properties that provide superior acoustic performance, especially in diced small elements within the TUT-array[50]. While PMN-PT inherently possesses optical transparency[51], high-precision lapping and polishing are applied to achieve a clear surface for enhanced transparency. Both sides are fully coated with indium-tin-oxide (ITO) as a transparent electrode, followed by gold stripes along the edges to improve electrical connection efficiency with electrodes of other layers. The first matching layer (ML1), made of a SiO₂-epoxy composite[39], is coated with gold applied as stripes along its top, bottom and side edges. The second matching layer (ML2) is made of Epotek 301. The PMN-PT and two MLs are cut with a dicing saw into rectangles measuring 23 × 5.5 mm in width and length. The TUT-array uses two flexible printed circuit boards (FPCBs): one has connectors for each channel control, and the other has a connector for the common voltage ground. The FPCBs feature a central opening, measuring 20 × 3.5 mm along the x and y axes, respectively, creating a physical void to allow the laser beam to pass through. A polycarbonate backing block (BB) is prepared, with its top surface having the same area as the PMN-PT rectangle. (2) After a small amount of bonding epoxy is applied between each layer, all of the layers are stacked in the bottom-to-top order of the BB, FPCB1, PMN-PT, ML1, FPCB2, and ML2. (3) The stacked assembly is pressed and heat-cured at 65 °C for 4 h, establishing firm physical contact between the layers. It is then cooled at room temperature for 24 h to ensure thermally stabilization. As a result, the gold stripe on the bottom surface of the PMN-PT is electrically connected to the gold on FPCB1, while the gold stripe on the top surface of the PMN-PT is electrically connected to the gold on FPCB2. (4) The bonded stack assembly is then diced using a 25 μm thick blade, forming 64 individual elements. (5) The diced stack is secured in a customized steel acoustic lens mold, then highly transparent silicone is poured, degassed, and cast into the mold. In a cross-sectional view cut through its center (Fig. 1b), all elements are precisely aligned at a 300-μm pitch. Figure 1c shows a photograph of the fully fabricated TUT-array at front- and back-side views. Notably, in the back-side view, the transparency of the US effective footprint in the TUT-array allows the background text to be clearly visible. Additionally, in the close-up image, the separated elements can be faintly distinguished.

### Acoustic and optical characteristics of the TUT-array

Prior to operating the TUT-array, we applied a high re-poling voltage (~180 V) to each element for 60 s to enhance dielectric performance, as the polarization of the PMN-PT may have degraded during fabrication, potentially compromising acoustic performance (Fig. S1, Supplementary information). To assess the acoustic performance, we used a pulse-echo test to examine the impulse response and verify the acoustic signal and its frequency response. The acoustic waveform was appropriately attenuated, with a duration of less than 1 μs, and the frequency range exhibited a center frequency of 7 MHz and a bandwidth of 45%, suitable for imaging purposes (Fig. 2a). The electrical impedance and phase angle spectra were measured to determine the electrical characteristics of the TUT-array across different frequencies (Fig. 2b). The resonance ($f_r$) and anti-resonance ($f_a$) frequencies were

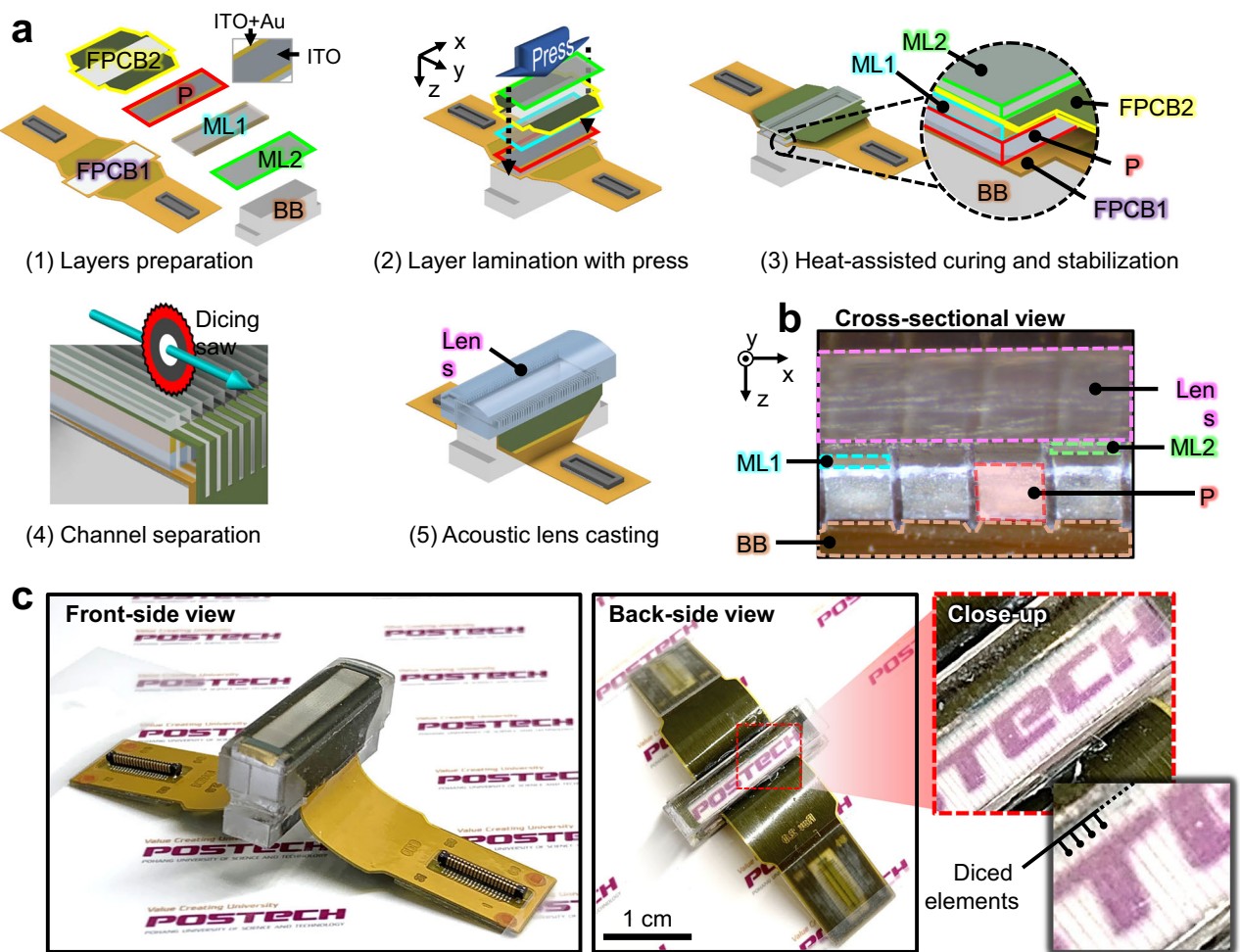

**Fig. 1 | Fabrication process of a TUT-array. a** A step-by-step overview of the TUT-array fabrication process. **b** Photograph of a cross-section of the TUT-array. All the elements are precisely aligned. **c** Photographs of the TUT-array in front- and back-side views. ML matching layer, FPCB flexible printed circuit board, P PMN-PT, BB backing block, ITO indium-tin-oxide, and Au gold.

observed as distinct peaks at 6.53 MHz and 7.54 MHz, respectively. The effective electromechanical coupling coefficient ($k_{eff}$) was calculated to be 0.5. These results were consistent with Krimholtz-Leedom-Matthaei (KLM) modeling-based simulations (Fig. S2, Supplementary information), confirming that the TUT-array had been fabricated as designed. In addition, the acoustic beam field generated by the TUT-array matched well with the simulation results (Fig. S3, Supplementary information). The averaged center frequency and bandwidth of all elements were calculated to be 7 MHz and 45%, respectively (Fig. 2c). Based on the pulse-echo waveform of each element, the averaged SNR was 45 dB (Fig. 2d). In addition, crosstalk measurements demonstrated the effectiveness of the appropriately spaced pitch and the damping effect of the filled silicone between the elements (Fig. 2e). The measured values were lower than the standard threshold of −30 dB in the 1–9 MHz frequency range, with −35.5 dB for adjacent elements and −41.1 dB for the second nearest elements.

Regarding the optical characteristics of the TUT-array, we spectroscopically examined each layer individually across the wavelength range of 400–900 nm. As seen in Fig. 2f, the near-infrared (NIR) optical wavelength region, widely used in PAI and FLI, is highlighted in a yellow box, and each TUT-array component exhibits light transmittance above 76.8% in this region. The fabricated TUT-array demonstrates a peak transmittance of 72.7% at 720 nm, with relatively constant optical transmittance across the NIR region. We compared the acoustic beam field of a commercial opaque UT (64

elements; center frequency of 8 MHz; bandwidth of 60%; L12-5A; and SonicLab Co., Ltd.) with that of the prototype TUT-array (Fig. 2g). Both the TUT-array and the commercial opaque UT operate central 32 elements, each with a driving voltage of 3.5 V. The US beam was focused at a depth of 25 mm, and detection was performed using a hydrophone. The SNR and acoustic pressure of the TUT-array were respectively 1 dB and 22 kPa lower than those of the commercial UT. Additionally, when measuring the full-width-at-half-maximum (FWHM) values of the acoustic fields, the TUT-array showed only a slight difference of 0.03 mm in the x-axis (lateral direction) and 0.14 mm in the y-axis (axial direction). These results indicate that the TUT-array's performance was comparable to that of the commercial UT, with only minimal differences.

## Performance benchmarking of triple-modal US/PA/FL imaging with an TUT-array opto-US probe

To investigate the potential of a triple-modal USI/PAI/FLI system using the TUT-array, we benchmarked the system's performance in phantoms (Fig. 3). A schematic of the multimodal opto-US probe is illustrated in Fig. 3a. Similar to conventional UTs, the handheld, compact, and user-friendly probe is designed to be easy for clinicians to adopt. The opto-US probe is composed of the TUT-array and an optical module. Within the optical module, light is delivered through an optical fiber and transformed into a line beam by an engineered diffuser (ED). The line beam is then reflected by the mirror and dichroic

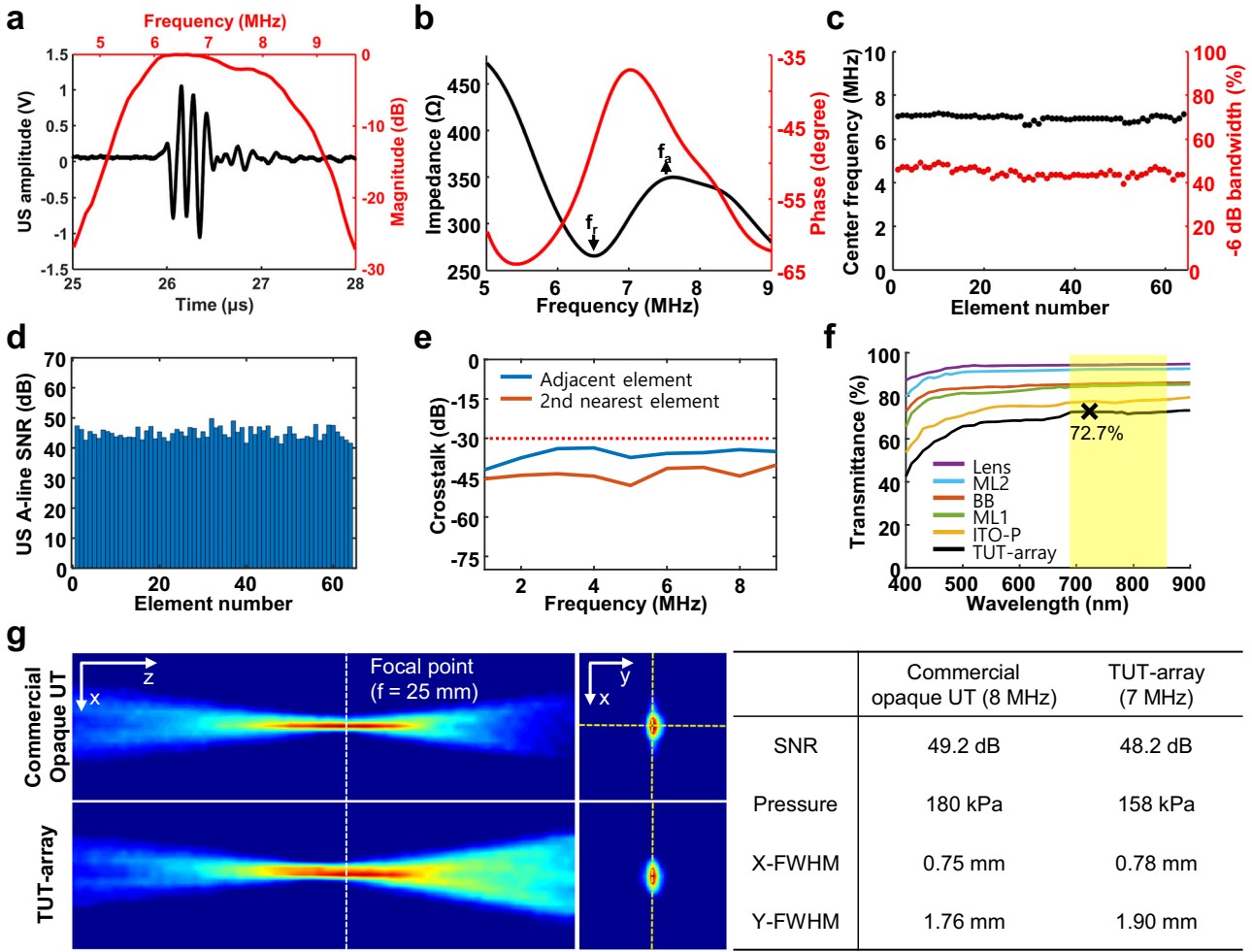

**Fig. 2 | Acoustic and optical characteristics of the TUT-array. a** Experimental pulse-echo impulse response waveform (black) and corresponding frequency spectrum (red) of the TUT-array. **b** Experimental electrical impedance (black) and phase angle (red) of the TUT-array. **c** Quantified center frequency and −6 dB bandwidth for each element. **d** Quantified ultrasound (US) A-line signal-to-noise ratio (SNR) for each element. **e** Crosstalk between adjacent and second nearest elements. **f** Light transmittance of layers composing the TUT-array. **g** Comparison of the acoustic beam fields of the commercial opaque US transducer and the TUT-array. ML matching layer, ITO-P ITO coated PMN-PT, BB backing block, and FWHM full-width-half-at-maximum.

mirror (DM), passing through the TUT-array and a US gel-pad to reach the target. The TUT-array is positioned coaxially at the end of the light path, enabling both the transmission and reception of US waves in a coaxial configuration. An NIR camera placed behind the DM captures a clear field of view (FOV) beyond the TUT-array for wavelengths above 800 nm. The photographs show the internal view of the opto-US probe and clearly illustrate how the laser illumination passes through when it is turned on and off. More detailed information is provided in the "Methods" section. Altogether, our new opto-US probe is much thinner, smaller, and lighter than the previously reported handheld PAUS probe with an opaque UT[24,52–55]. These improvements result from both the coaxial alignment of the acoustic and optical planes, and the significantly smaller laser delivery unit, achieved by replacing a bulky fiber bundle with a single optical fiber, which further enhances the optical coupling efficiency.

The opto-US probe is connected to a laser and data acquisition (DAQ) system, establishing an integrated triple-modal imaging system (Fig. S4, Supplementary information). The US Tx/Rx system is programmable for each TUT-array element, enabling US transmission and reception along with the detection of laser-induced PA signals. Additionally, FL images are acquired in real-time through the USB-connected NIR camera, delivering *en-face* visualization directly to the computer. This setup streamlines the triple-modal USI/PAI/FLI imaging

acquisition by following a synchronized timing sequence to ensure comprehensive imaging across modalities (Fig. S5, Supplementary information). To assess the triple-modal USI (Fig. 3b, c), PAI (Fig. 3d), and FLI (Fig. 3e) system's performance with the opto-US probe, we conducted imaging experiments in phantoms for each modality. Initially, we focused the US transmission at a depth of 20 mm. For USI, we imaged line targets (Fig. 3b) and anechoic targets (Fig. 3c) using a standard US phantom model. Based on Fig. 3b, the USI lateral and axial resolutions were measured to be 0.31 mm and 0.39 mm, respectively, with the lateral resolution gradually decreasing as the depth increases (Fig. 3f). Additionally, as shown in Fig. 3c, the US contrast-to-noise ratio (CNR) was calculated to be 7.2 at a depth of 20 mm (Fig. 3g). The lateral and axial resolutions of our TUT-array are almost identical to those of the commercial opaque UT, and its CNR, measured using a grayscale target, shows an average difference of only 0.17 dB (Fig. S6, Supplementary information). Thus, the TUT-array shows performance quite comparable to that of the commercial opaque UT. To evaluate the PAI performance, we prepared a tissue-mimicking phantom with black nylon strings[56] and acquired its cross-sectional image (Fig. 3d). The PA spatial resolution was measured from the string's profile, yielding lateral and axial resolutions of 0.38 mm and 0.71 mm, respectively (Fig. 3h). The PAI performance of the opto-US probe was compared to that of the commercial UT with an oblique laser alignment, showing

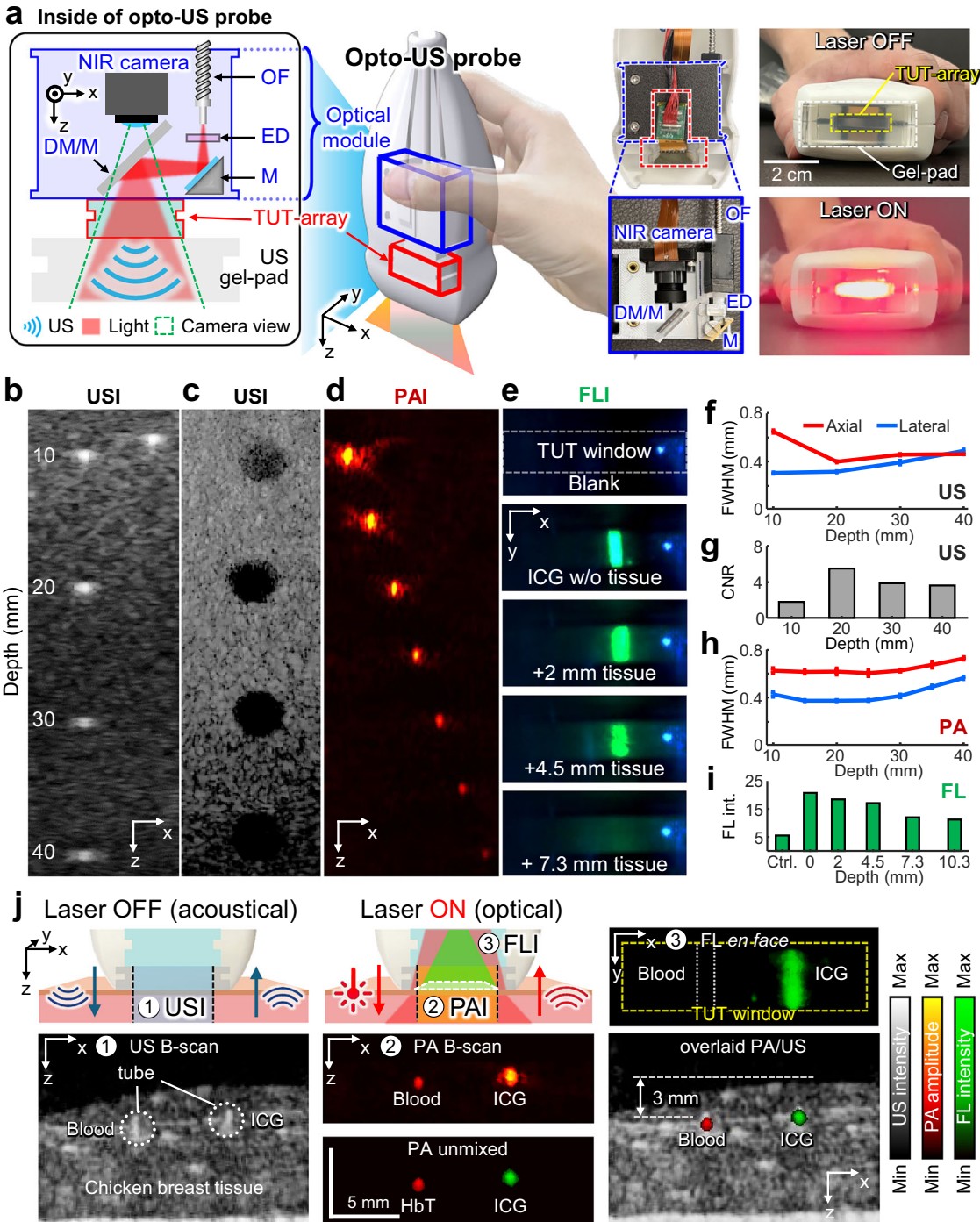

**Fig. 3 | Triple-modal USI/PAI/FLI imaging system with an opto-US imaging probe via a TUT-array and its performance benchmark. a** Schematic and photographs of the opto-US probe, showing all optical and US components. The laser beam passes directly through the TUT-array, illustrating its transparency, which enables the beam to be coaxially aligned with the US transmission path. USI, PAI, and FLI phantom imaging performance via the opto-US probe. **b** US B-scan images with wire targets and **c** anechoic targets in a standard US phantom. **d** PA B-scan image with black strings in a tissue-mimicking phantom. **e** FL en-face image of a tube filled with indocyanine green (ICG) beneath stacked layers of chicken tissue. **f** USI spatial resolution. **g** USI contrast-to-noise ratio (CNR). **h** PAI spatial resolution. **i** FL intensity vs depth. **j** Simultaneous trimodal USI/PAI/FLI of tubes filled with blood and ICG beneath stacked layers of chicken tissue. DM dichroic mirror, M mirror, ED engineered diffuser, and OF optical fiber.

only a slight lateral resolution difference of 0.03 mm and an axial resolution difference of 0.07 mm, presumably arising from the slightly lower center frequency of the TUT-array (7 MHz) than the commercial UT (8 MHz). However, the opto-US probe with coaxial laser alignment achieved an average of 7 dB higher SNR with depth, indicating improved efficiency in PAI (Fig. S7, Supplementary information). While Fig. S7 shows comparable SNR degradation between the TUT-array and

the commercial probe, this result reflects both the acoustic sensitivity and the light delivery efficiency. To clarify the specific impact of coaxial versus oblique illumination, we performed an additional phantom experiment using the same TUT-array under both configurations, demonstrating that coaxial illumination yields a higher SNR of 8 dB and more uniform signal decay with depth (Fig. S8, Supplementary information). Moreover, the opto-US probe can

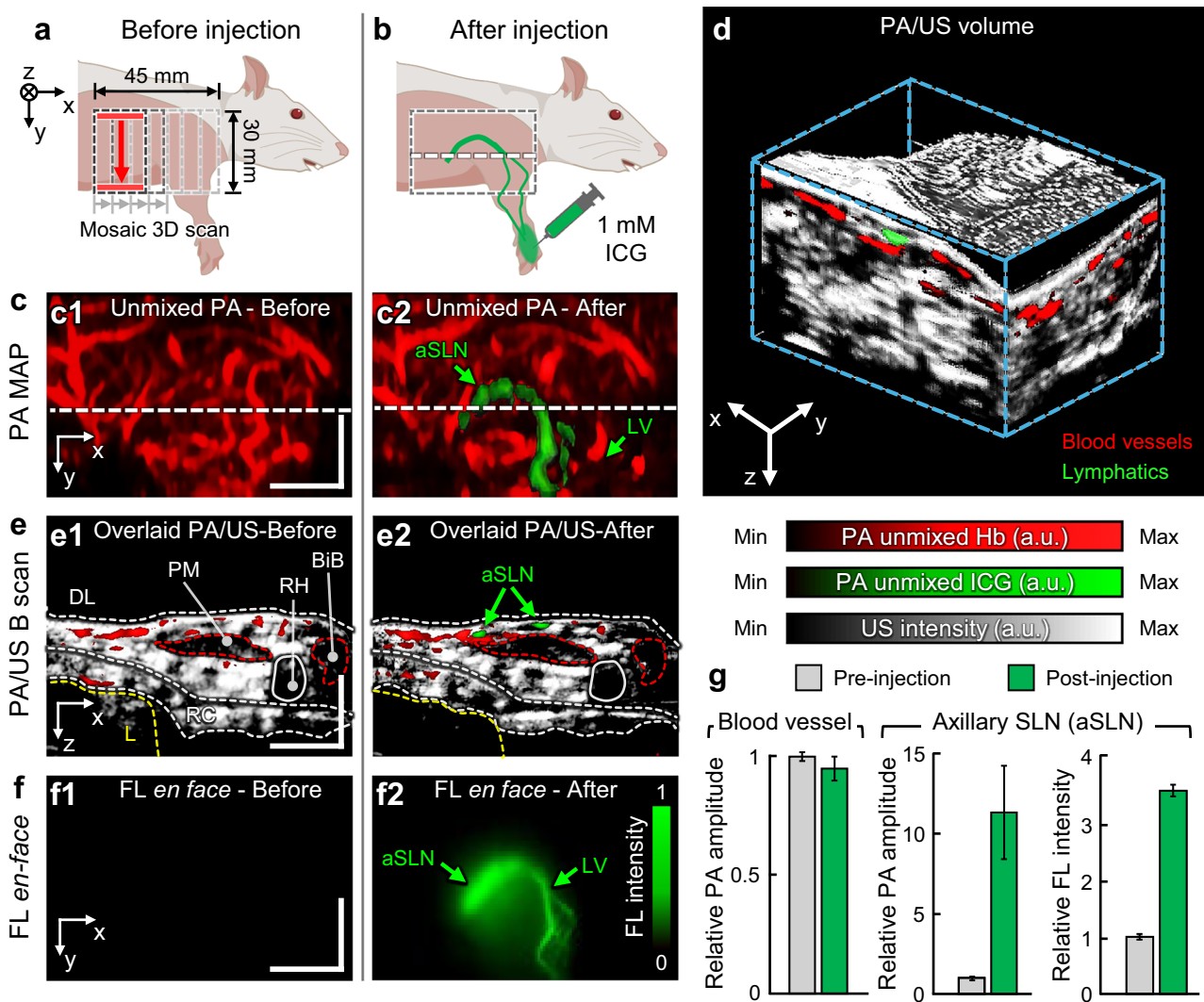

**Fig. 4 | 3D triple-modal mapping of lymphatics in a live rat.** Simultaneous ultrasound (US)/photoacoustic (PA)/fluorescence (FL) imaging was performed via mosaic 3D motorized scanning over the right axillary region of an anesthetized rat **a** before and **b** after indocyanine green (ICG) injection. **c** Maximum amplitude projection (MAP) of PA volume visualizing the vascular network across the sagittal plane, **c1** before and **c2** after the ICG injection. The path of the overlaid unmixed ICG contrast (in green) in (**c2**) visualizes the axillary lymphatics, featuring a continuous connection from peripheral lymphatic vessels (LV) to the axillary sentinel lymph node (aSLN). **d** Coregistered PA/US 3D volume over the axillary region

(Supplementary Movie 1). LV and superficial veins are identified underneath the dermal layer (DL). **e** Overlaid PA/US B-mode images **e1** before and **e2** after the ICG injection. Detailed musculoskeletal structures and organ anatomy are distinguishable from USI, while PAI demonstrates branching blood/lymphatic vessel distribution. **f** Mosaic-synthesized FLI via the TUT optical window **f1** before and **f2** after ICG injection. **g** Quantified PA and FL signal enhancement with ICG injection (mean ± standard error, *n* = 3, independent experiments). PM pectoral muscle, RH right humerus, BiB biceps brachii, RC rib cage, L liver. Scale bar = 1 cm. All values are expressed in a.u., which denotes arbitrary units.

simultaneously provide FLI from the top view, utilizing the TUT as an optical window. To benchmark the FLI performance, we imaged an ICG-filled tube beneath stacked layers of chicken tissue (Fig. 3e). The ICG-filled tube remained clearly visible with high intensity up to a depth of 4.5 mm and detectable up to a maximum depth of 7.3 mm, both with and without the TUT-array (Fig. 3i). The FL SNRs in both configurations, where the average SNR difference between measurements with and without the TUT-array is 0.82 dB across various depths (Fig. S9, Supplementary information).

To demonstrate simultaneous USI/PAI/FLI, two tubes, one filled with mouse blood and the other filled with ICG dye, were placed beneath chicken tissue (Figs. 3j and S10, Supplementary information). For multispectral PAI, we used 690, 720, 756, and 796 nm wavelengths. In the PA B-scan image, both blood and ICG are well visible, with the ICG signal appearing stronger than the blood signal. A spectral unmixing algorithm, isolated and visualized the ICG signal. Although USI alone could not distinguish between blood and ICG, a PA/US

overlaid image did provide depth information for blood and ICG, which were measured to be 3 mm deep below the skin. In addition, the FLI *en-face* image offered an intuitive top view of the location of the ICG.

## 3D triple-modal mapping of lymphatics in live animals

To investigate the system's trimodal imaging capability in vivo, we performed simultaneous PA/US/FL imaging to map the lymphatics of an anesthetized Sprague Dawley rat (Fig. 4). To volumetrically scan the rat's entire right upper limb region, we attached the opto-US probe to a motorized stage and performed mosaic mechanical raster scanning to create a 45 × 30 mm² synthetic aperture from five overlapping linear scans (Fig. 4a). In each scan slice, we acquired PA B-mode images at 690, 720, 756, and 790 nm (Fig. S11, Supplementary information), one US B-mode image, and one FL image at 756 nm. These images were stacked to create synthetic 3D PA/US volumes and FL *en-face* image. After preinjection imaging, a 0.2 mL of 1 mM ICG solution was injected

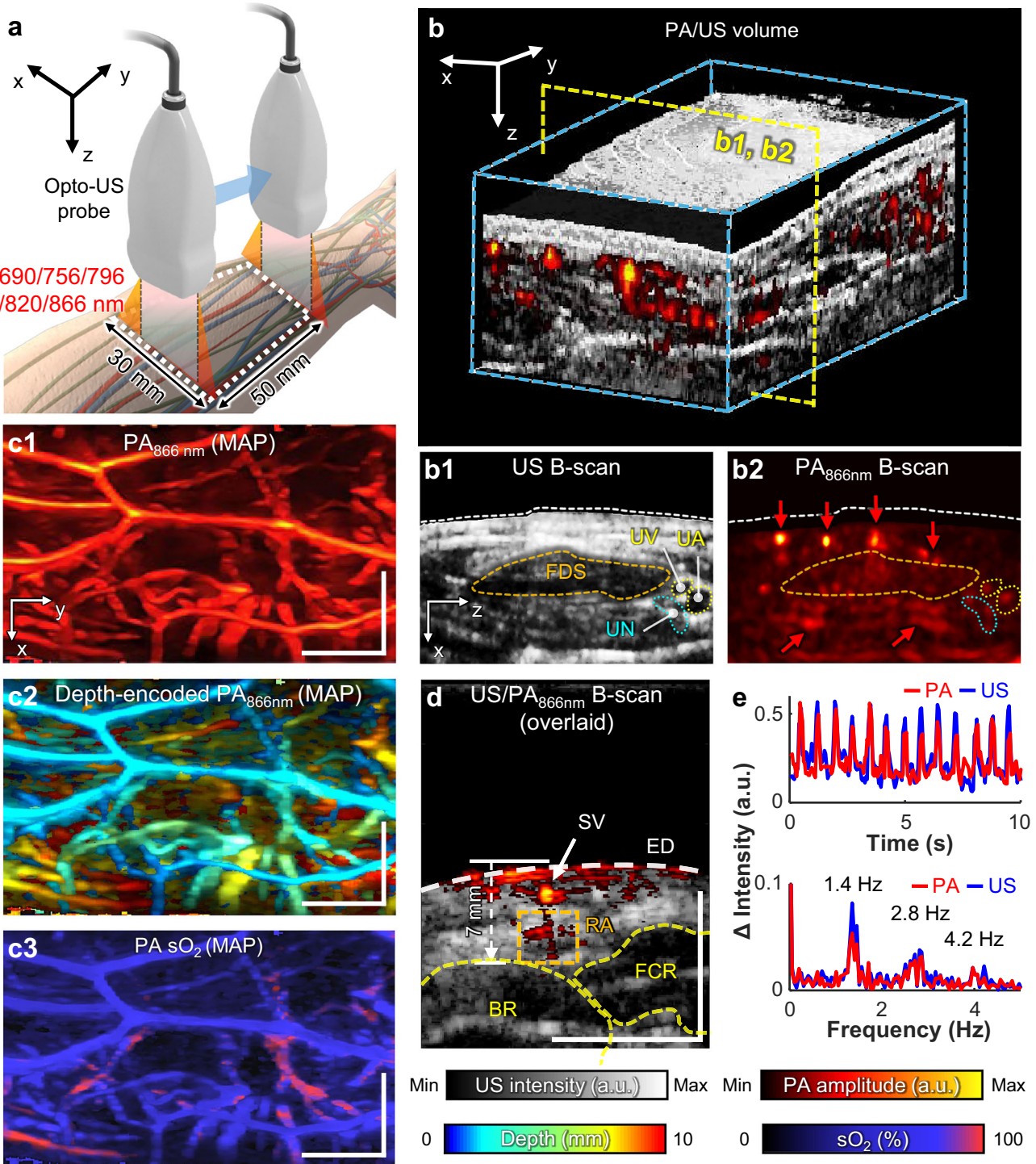

**Fig. 5 | Bimodal 3D functional PA/US imaging of a human arm in real time.**
**a** Experimental configuration. **b** 3D rendering of overlaid PA/US volumes, simultaneously describing the tissue anatomy from US and registered vasculatures in PA acquired at 866 nm (PA866 nm) in the forearm (Supplementary Movie 2). Representative B-mode US (**b1**) and PA (**b2**) depict distinguishable musculoskeletal structures and adjacent blood vessels, respectively. **c1**–**c3** Maximum amplitude projection (MAP) in the x-y plane for **c1** PA866nm, **c2** depth encoded PA866nm, and **c3** oxygen saturation (sO2) map distinguishes the arterial network from the superficial venous network. **d** B-mode US/PA image cut along the dotted line in (**c1**). Pulsatile arterial motion identified from the B-mode US/PA images (Supplementary Movie 3). **e** The temporal change and corresponding frequency spectrum demonstrate 1.4 Hz (84 bpm) pulsation in both US and PA images in the arterial region-of-interest (ROI) marked with a yellow box in (**d**). UA ulnar artery, UV ulnar vein, UN ulnar nerve, FDS flexor digitorum superficialis, SV superficial vein, ED epidermis, RA radial artery, BR brachioradialis, and FCR flexor carpi radialis. Scale bars = 1 cm. All values are expressed in a.u., which denotes arbitrary units.

intradermally into the right frontal paw, followed by postinjection imaging over the same FOV (Fig. 4b).

Both PA volumes consistently showed abundant vessels extending into the right lateral region (Fig. 4c). The pre-injection PAI revealed

subcutaneous blood vessels spanning the right forelimb and thoracic region (Fig. 4c1). The post-injection PA image shows a similar vessel distribution, but with significant local PA signal enhancement from the right frontal limb down to the shoulder. After spectral PA unmixing,

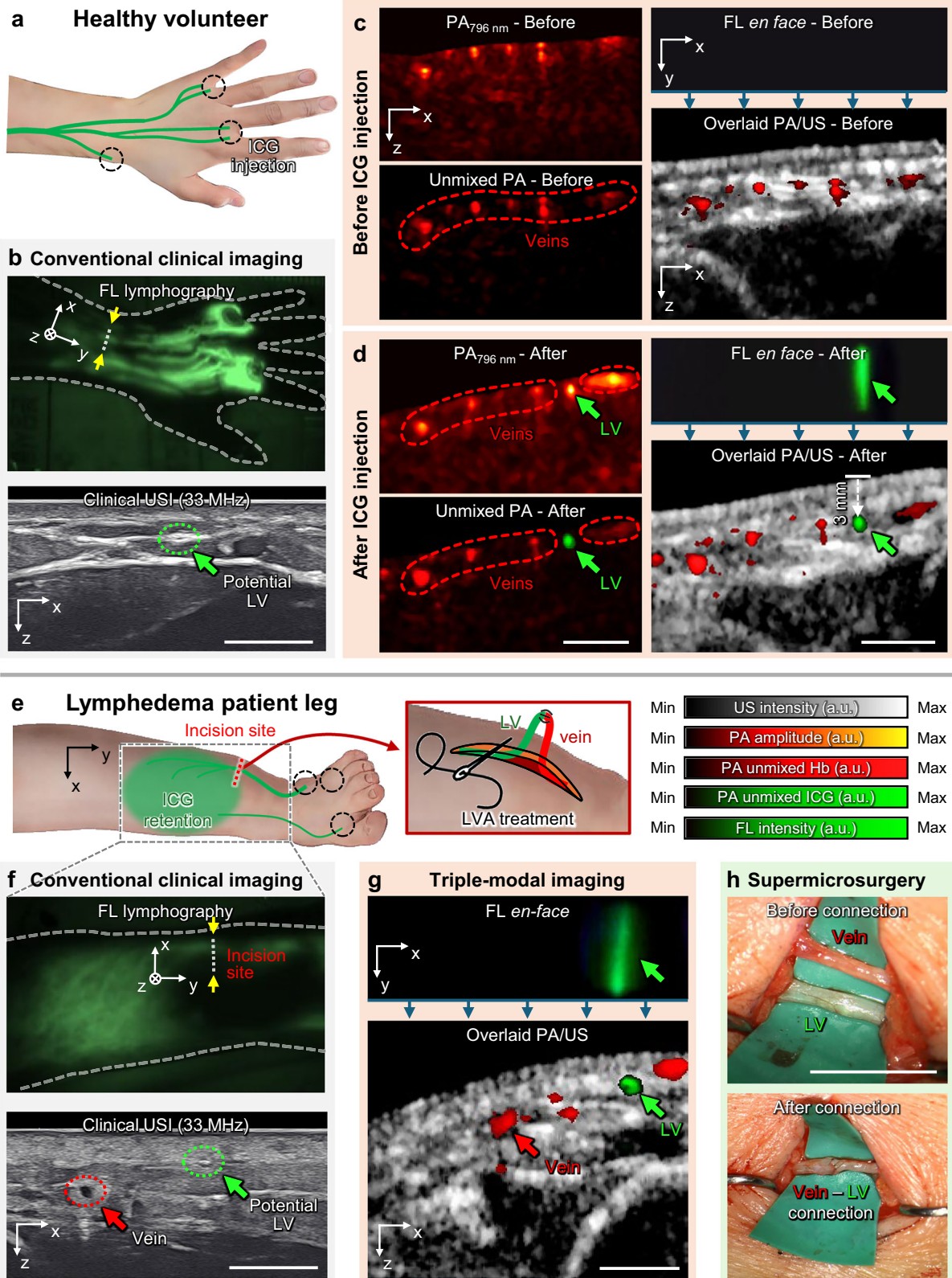

**Fig. 6 | Triple-modal USI/PAI/FLI guided microsurgery to perform lymphaticovenous anastomosis (LVA), of lymphedema patients. a** Lymphography of a healthy volunteer with indocyanine green (ICG) injection (Supplementary Movie 4). **b** Conventional clinical imaging methods: fluorescence (FL) lymphography and ultrasound imaging (USI, 33 MHz). Red and green arrows indicate veins and potential lymphatic vessels (LVs), respectively. Triple-modal USI/PAI/FLI using the opto-US probe **c** before and **d** after ICG injection. **e** Preoperative lymphatic illustration and ICG injection sites in the leg of a lymphedema patient (Supplementary Movie 5). **f** Conventional clinical imaging examination. **g** Triple-modal USI/PAI/FLI examination of LVs and blood vessels. **h** Photographs acquired before and after LVA supermicrosurgery. Scale bar = 5 mm. All values are expressed in a.u., which denotes arbitrary units.

the stream of unmixed ICG reveals the lymphatic network in the right upper limb, flowing from the periphery and toward the thorax (Fig. 4c2). The axillary sentinel lymph node (aSLN) appears as a major node where the ICG stream aligns following the axillary curve. In the co-registered PA/US volume, US imaging visualizes the anatomical structures of major organs and musculoskeletal features in the right scapular and thoracic regions, while PA imaging identifies sub-cutaneous blood and lymphatic vessels traversing the connective tissues (Supplementary Movie 1 and Fig. 4d, e). Notably, the lymphatic vessels are invisible in the US images and are distinguishable only in the PA images, where they are located approximately 3 mm beneath the dermal layer, alongside the adjacent superficial veins[52]. ICG serves as both a PA contrast agent and a fluorophore, allowing the lymphatic network to also be visualized in the corresponding FL images, where it exhibits a high spatial correlation with the vascular morphology in the PA lymphatic maps (Fig. 4f). The PA and FL signal enhancement after ICG injection can be further quantitatively assessed from the imaged major blood vessel and aSLN (Fig. 4g), demonstrating a mere difference from blood vessel while significant signal increase at the aSLN region, measured as 11-fold and 3.6-fold increase from PAI and FLI, respectively. Overall, the equivalent discoveries of the lymphatic network in both PAI and FLI, along with the simultaneous trimodal imaging that combines anatomical structure and molecular imaging, successfully demonstrates the seamless integration of optical and acoustic imaging using the TUT.

### Clinical 3D functional PA/US vascular imaging of a human arm

To investigate the system's clinical feasibility, we employed the opto-US probe for 3D functional PA/US vascular imaging of a human arm (Supplementary Movie 2 and Fig. 5a). As in the preclinical imaging, we performed 3D mosaic scanning of a $30 \times 50\,mm^2$ area of a healthy volunteer's forearm, capturing two lanes of overlapping linear scans. We performed multispectral PAI with 690, 756, 796, 820, and 866 nm laser excitation to precisely unmix blood contrast into deoxygenated (Hb) and oxygenated hemoglobin ($HbO_2$) molecular contrast. Because we were not performing FLI, the NIR camera and dichroic mirror inside the optical module were replaced with a standard mirror to utilize a broader band of laser excitation. The acquired US/PA volume data were synthetically merged into a wide FOV, resulting in a 3D image volume as shown in Fig. 5b.

The US (Fig. 5b1) and PA (Fig. 5b2) images provide detailed cross-sectional views of upper limb tissue structures (such as the ulnar artery, vein, nerve, and flexor digitorum superficialis) and adjacent blood vessels, enabling precise anatomical localization and assessment of vascular and tissue morphology. In the PA MAP image, the intricate vascular network within the human arm is clearly visualized in a top-projection view (Fig. 5c1). We further generated a depth-encoded MAP image conveying the depth information of PA contrasts with pseudo-colors, which clearly illustrates the vascular connectivity at depths ranging from 0 to 10 mm (Fig. 5c2). Additionally, $sO_2$ levels calculated from the ratio between the unmixed $HbO_2$ to the total Hb (the sum of the unmixed Hb and $HbO_2$) are spatially mapped over the PA MAP image, displaying highly oxygenated vessels in red and less oxygenated vessels in blue (Fig. 5c3). The $sO_2$ is measured with high consistency along the vascular continuity, highlighting a number of vessels in the deep region with an approximately 90% $sO_2$ level. We speculate that these vessels are arteries, located in relatively deeper regions than superficial veins and containing richly oxygenated blood.

Beyond its multispectral imaging capabilities, the 20-Hz frame rate of the bimodal PA/US imaging captures cardiac circulation dynamics (Supplementary Movie 3 and Fig. 5d, e). We continuously imaged a forelimb cross-section over 10 s (200 frames) at a single 866 nm wavelength (Fig. 5d). Pulsatile motion was clearly observed at an identical location in both the US and PA images. To quantitatively assess this motion, we defined a region of interest (ROI) encompassing

the arterial lumen and walls, and measured the frame-by-frame intensity fluctuations from both imaging modalities (Fig. 5e). The pulsation patterns from the US and PA images were temporally correlated, exhibiting synchronized systolic peaks. Spectral analysis revealed multiple harmonic peaks with a fundamental frequency at 1.4 Hz (84 bpm), confirming the regular heartbeat of the healthy volunteer.

### Lymphaticovenous anastomosis (LVA) microsurgery of lymphedema patients guided by triple-modal USI/PAI/FLI

A LVA procedure is a minimally invasive microsurgery connecting LVs (diameter <0.5 mm if normal, diameter <1 mm if lymphedematous, depth <5 mm) to nearby veins (diameter ~0.5 mm, depth <1 cm) to facilitate lymph drainage[57,58]. After ICG injection, surgeons use both FLI and USI to preoperatively confirm the location of veins and LVs, FLI captures the ICG stream in the top view, while USI offers depth-sensitive location information of the LVs and veins for microsurgery. However, pinpointing LV locations becomes challenging in severe lymphedema patients where FLI lymphatic contrast is only faintly visible or disrupted by abnormal patterns due to ICG retention, and USI lacks of specificity in distinguishing between veins and ICG-dyed lymphatic vessels. In these typical cases, operators often need to repetitively use two separate FLI and USI systems with different FOVs, which complicates and prolongs the final confirmation and relies on the operator's empirical judgment to identify lymph vessels.

In contrast, the opto-US probe is particularly well-suited for guiding LVA procedures, since it simultaneously provides comprehensive triple-modal US/PA/FL guided images within a single device. Before applying the triple-modal imaging system to patients, we demonstrated its capabilities in healthy volunteers (Fig. 6a–d and Supplementary Movie 4). ICG dye was intradermally injected via the interdigital web of the volunteer's hand (Fig. 6a). The normal perfusion of ICG through the LV network was subsequently confirmed using commercial FL lymphography (Fluobeam 800, Fluoptics, France), while the FL trace of the LVs from the top view facilitated transcutaneous inspection using a 33 MHz high-frequency clinical USI system (Aplio i800, Canon Medical Systems, Japan) (Fig. 6b). High-frequency USI enabled identification of LVs based on their shape, echogenic texture, and response to pressure, however, differentiating LVs from adjacent veins remained challenging and time-intensive. Next, using the opto-US probe, we demonstrated its effectiveness in identifying LVs and adjacent veins both before and after ICG injection. Before the ICG injection (Fig. 6c), multispectral PAI was performed, acquiring strong PA signals in the transcutaneous layer. Spectral PA unmixing allowed visualization of the Hb distribution, while the FL en-face image showed no detectable signal, as expected before ICG injection. The overlaid PA/US image confirmed the location of veins within the transcutaneous layer. After ICG injection, triple-modal images were acquired at the same location (Fig. 6d). Spectral PA unmixing distinctly separated Hb and ICG signals, precisely localizing LVs and blood vessels. The FL en-face image also identified LVs at the same lateral position as the unmixed PA ICG image. Furthermore, the overlaid PA/US image confirmed that the LVs were located approximately 3 mm beneath the skin surface, along with surrounding blood vessels.

We then demonstrated the clinical feasibility of the triple-modal imaging system in determining surgical incision sites in two lymphedema patients scheduled for LVA treatment (one lymphedema patient in Fig. 6e–h, the other in Fig. S12, and Supplementary Movie 5). The opto-US probe's simultaneous triple-modal imaging capability enabled non-invasive identification of both surgically joinable LVs and available veins, potentially allowing for minimal incisions. A day before surgery, patients received ICG injections in the interdigital web and dorsum of the foot (Fig. 6e). An FL lymphography image showed typical FL contrast with a diffuse cutaneous reflux pattern, as well as a faint trace of the distal LV adjacent to the lateral ankle (Fig. 6f). Surgeons had

marked potential incision sites along clear LV streams before diffusion; however, revisiting these locations with clinical USI proved challenging due to the difficulty in distinguishing LVs from blood vessels or nerves, resulting in only potential LV contrasts being marked. In contrast, the triple-modal imaging effectively differentiated these vessels into veins and LVs. In the unmixed PA images, molecular contrasts clearly distinguish LVs (ICG) from adjacent veins (Hb) beneath the skin layer, spatially coinciding with the LV-specific contrast in the *en-face* FL image (Fig. 6g). Among numerous candidate veins, the surgeon selected a vein having a similar size (~1 mm diameter) and depth (3 mm depth below epidermal layer) to those of the LV. During the supermicrosurgery, the surgeon made a 5 mm lateral minimal incision at the predetermined site and identified both a 0.9 mm thick LV and a 0.8 mm-thick vein at a depth of 3 mm (Fig. 6h). Knowing the thickness, depth, and relative positions of both LVs and veins from preoperative triple-modal imaging is particularly important in helping the surgeon avoid repeated incisions and laborious manual vein discovery, which typically increases the risk of lymphedema recurrence and need for reoperation. In this demonstration, the surgeon successfully conducted the LVA procedure with reduced operation time and a minimized incision size. In summary, the triple-modal capability of the opto-US probe has proven to be an intuitive and time-efficient clinical solution for lymphatic-vein mapping prior to LVA treatment.

## Discussion

Multimodal imaging systems combining OI and USI are increasingly valued for providing comprehensive diagnostic information and therapeutic monitoring by leveraging the complementary strengths of each modality. However, due to the opacity of conventional UTs, integrating these modalities often increases system complexity. The TUTs could solve this issue, but existing single-element designs are unsuitable for real-time clinical imaging. In this study, we developed a linear TUT-array (64 elements) with a 7 MHz center frequency, 45% bandwidth, and 72.7% optical transmittance. The TUT-array was successfully integrated with OI, including PAI and FLI, into a handheld opto-US probe. This compact opto-US probe displays USI, PAI, and FLI simultaneously, providing the anatomical structure of the tissue, microvascular networks, and exogenous agent-containing tissues.

The imaging experiments conducted with the TUT-array in this study demonstrated a comparable SNR for USI, while its transparency allowed for seamless integration of PAI and *en-face* FLI. These advantages enable complete coaxial integration of acoustic and OI modalities, specifically overcoming two key limitations of conventional handheld PAI. First, the coaxial arrangement improves the SNR and penetration depth by ensuring the shortest optical path (Fig. S13, Supplementary information). Optical fluence is exponentially attenuated in biological tissue; thus, the linear distance between the point of incidence and the target significantly affects PA SNR. With conventional handheld PAI probes, the angled optical path of the oblique illumination naturally creates a shallower penetration depth, whereas with coaxial illumination, the perpendicular incidence allows for deeper and more efficient light delivery into the tissue. Second, this coaxial alignment promotes complete coherence between the optical and acoustic planes throughout the full imaging depth (Fig. S14, Supplementary information). In an oblique illumination setup, the optical plane intersects the acoustic plane at only a single depth, leading to depth-dependent variations in PA yield due to misalignment. Consequently, oblique configurations are particularly susceptible to depth variation when scanning curved or irregular surfaces. In contrast, the TUT-array's coaxial illumination maintains consistent alignment between the optical and acoustic planes across all depths, providing more uniform imaging across various tissue geometries.

Integrating multiple imaging modalities into a single opto-US probe offers significant clinical advantages. Comprehensive validation from both in vitro imaging and preclinical and clinical in vivo imaging

demonstrates that simultaneous triple-modal imaging provides relatively effortless visualization of the lymphatic and vascular network, compared to conventional methods that alternate between clinical lymphography and USI. In particular, multispectral PAI provides distinct molecular contrasts for ICG and Hb, serving as a cross-validative modality that bridges the identification of LV streams in FLI with tissue structure visualized from USI. As described here, convenient localization of lymphatic and blood vessels with real-time preoperative imaging aided in planning minimal incisions for LVA in two lymphedema patients, effectively reducing time and costs for both surgeons and patients. Our compact, handheld probe system can intraoperatively enable surgeons to probe LVs on-site in confined surgical environments and to adapt to different surgical scenarios. In sharp contrast, previously reported PA lymphography[48] does not support USI/FLI and requires considerable time for visualization and substantial dedicated space, making it far less suitable for intraoperative use.

Despite its strong performance, the TUT-array's 64-channel configuration limits its imaging range. Expanding to 128 channels and beyond could double the FOV, enhancing clinical utility[59]. However, scaling up poses significant fabrication challenges, particularly in aligning and bonding the transparent multilayer stack over a larger area. As the array size increases, it becomes more difficult to maintain uniform thickness, ensure consistent electrical connections, and preserve optical transparency with a smooth surface finish. To address these issues, precise control of such bonding parameters as pressure, adhesive distribution, and substrate flatness is required. In addition to high-resolution alignment, surface treatment methods such as plasma activation or precision polishing may also be required. Additionally, improving the conductivity of the transparent electrodes is essential, as the current ITO-coated electrodes show lower sensitivity than commercial UTs. Metal-based transparent electrodes offer better conductivity[60] but lack durability in mechanical processes like dicing. Further material optimization could improve the probe's performance.

Acknowledging the complementary strengths of PA and US imaging, the implementation of advanced functional techniques holds strong potential to enhance the diagnostic utility of this dual modality system, as demonstrated by multispectral PA imaging in this study. For instance, integrating Doppler US imaging could enable simultaneous assessment of blood flow and oxygenation, providing a pathway toward real time comprehensive metabolic rate imaging. Future work should focus on addressing technical challenges, particularly the need to accommodate the full Doppler US sequence, including successive acquisitions, image reconstruction, and clutter filtering within the laser pulse intervals. These challenges may be addressed through tailored optimization of acquisition parameters and GPU based parallel computing.

In conclusion, we developed a TUT-array that enables broad-wavelength light transmission, and we seamlessly integrated it into a handheld opto-US probe. The probe provides real-time triple-modal USI, PAI, and FLI, delivering valuable diagnostic information about acoustic reflection, optical absorption, and FL. This study marks the clinical application of multimodal imaging using the TUT-array for lymphedema patients. The opto-US probe demonstrates significant potential as a tool for guiding lymphedema surgery and could be expanded to other clinical applications. Based on these results, we firmly believe that triple-modal USI/PAI/FLI using the opto-US probe can be expanded to a wider range of clinical applications, beyond lymphedema treatment.

## Methods

### Re-poling process
Re-poling was conducted to restore the polarization of the elements, which may have been degraded by thermal and mechanical stresses during bonding and dicing. Re-poling was performed after the TUT-array was completely assembled, including electrical connections via

FPCBs and external cables. This step was particularly necessary due to the use of single-crystal materials and the complex bonding procedures required for optical transparency. A DC high-voltage supply (Kistech, Republic of Korea) was connected to each element through the signal and ground lines of the cable. To maintain the original polarization direction, a positive bias was applied to the ground line and a negative bias to the signal line. A high voltage (-180 V) was applied for 60 s per element under ambient conditions in air. Capacitance was measured before and after re-poling to assess changes in dielectric properties and confirm electrical recovery.

## TUT-array property measurements

The US impulse response was measured on a steel plate using a pulser/receiver (5800PR, Panametrics, USA), and the 64 channels were switched through a multiplexer. The electrical characteristics were analyzed with an impedance analyzer (E4990A, Keysight, USA). Crosstalk was evaluated by applying a sinusoidal signal to a single element and recording the resulting signals in adjacent and second-adjacent elements. The acoustic beam profile was captured using a needle-type hydrophone (NH500, Precision Acoustics, UK) and a measurement system (IPB760, IMP Systems, Republic of Korea). Alignment was facilitated by a 5-axis stage system. Beam profiles were taken point-by-point as the hydrophone moved across the field while 32 channels of the TUT-array transmitted signals. The measurement system enabled the evaluation of the SNR, acoustic pressure, and FWHM. The optical transmittance of the acoustic lens, matching layers, ITO-coated PMN-PT, and the fully assembled TUT-array were measured across the 400 nm to 900 nm range using a UV–VIS spectrophotometer (Cary 60 UV–Vis, Agilent, USA).

## Implementation of the opto-US probe

The Opto-US probe is composed of a TUT-array and an optical module, both of which are precisely aligned coaxially through the elaborate structure of the 3D printed housing. The TUT-array is interfaced with a US acquisition system (Vantage 256, Verasonics, USA), allowing for concurrent acquisition of US and PA signals. Within the optical module, a 1 mm core multimode fiber (OPTIBASE Inc., Republic of Korea) delivers a 20 Hz NIR laser beam (PhotoSonus M-20, Ekspla, Inc., Lithuania). The laser beam emitted from the fiber is reshaped into a line-shaped beam using an ED (EDL-40, VIAVI Solutions, USA) and reflected by a gold-coated glass mirror. The NIR laser beam is then reflected by a DM (#86-336, Edmund Optics, USA) and directed towards the imaged object. Note that the DM can be temporarily replaced with a typical mirror when using PAI only. For FLI, the ICG emission signal, which occurs above 800 nm, passes through the dichroic filter and is captured by a mini-camera equipped with a custom-designed 800–900 nm wavelength filter. To ensure smooth US signal transmission, a US gel-pad (BLUEMTECH, Republic of Korea) is installed in the housing as an acoustic couplant. The TUT-array is connected to a Vantage256 system (Verasonics, USA) via a custom-made cable, facilitating both US transmission/reception and PA signal acquisition. The FL images captured by the mini-camera are transferred to the Vantage256 computer via a connected USB.

## Real-time trimodal USI, PAI, and FLI sequence

To perform trimodal imaging acquisition with the handheld opto-US probe, we developed an imaging sequence externally synchronized to the laser source pulse repetition rate, as illustrated in Fig. S5, Supplementary information. The imaging sequence initiates PA acquisition for every instance of 20-Hz laser excitation, passively receiving the acoustic channel data without pulse transmission. Each PAI event is followed 100 us later by USI events involving three packets of 7 MHz line beam transmissions, steered in three angles (0°, −10°, +10°). Each packet consists of 64-line beams sweeping the FOV from left to right. Then the acquisition mode switches back to PA acquisition after finishing the last US transmit-and-receive event, waiting for the next laser trigger shot input. In the transition period, the receive gains are digitally switched to 54 dB and 39 dB for PA and US acquisition, respectively, to compensate for the relatively low PA receive signal level compared that of US (due to its acoustic passivity). Then, using time delay-based beamforming, the acquired PA and US channel data are both reconstructed into cross-sectional B-mode images with $40 \times 20$ mm FOVs in real time. For multispectral PA imaging, a pair of PAI/USI events is repeated by several wavelengths of a spectral cycle, at four wavelengths (690, 720, 756, 796 nm) for in vitro or five wavelengths (690, 756, 796, 820, 866 nm) for in vivo imaging. The latter configuration is optimized for sO2 mapping by including 820 nm and 866 nm, which are located above the isosbestic point at 796 nm, and by excluding 720 nm due to its spectral redundancy between 690 and 756 nm. The NIR camera keeps records of the FL contrast for ICG for the whole course of the experiment, and the images at 756 nm excitation are exclusively extracted with off-line processing. For 3D scanning, the US platform triggers the external motor stage system after the last set of US events, which translates the probe 0.5 mm in the elevational direction.

## Spectral unmixing algorithm

Spectral unmixing of each PA B-scan image was conducted using the following procedure[53,61–63]. To address the attenuation of optical fluence with increasing tissue depth, the background signal was determined at each depth, and the fluence was then normalized based on these calculated values. The multispectral PA images, after fluence compensation, were spectrally unmixed into the components of oxyhemoglobin, deoxy-hemoglobin, and ICG using a pseudo-inverse matrix method:

$$[C_1 C_2 \cdots C_n] = [PA_1 PA_2 \cdots PA_k] \cdot M^T \cdot [M \cdot M^T]^{-1} \tag{1}$$

$$M = \begin{bmatrix} \mu^1_{\lambda_1} & \mu^1_{\lambda_2} & \cdots & \mu^1_{\lambda_k} \\ \mu^2_{\lambda_1} & \mu^2_{\lambda_2} & & \mu^2_{\lambda_k} \\ \vdots & & \ddots & \vdots \\ \mu^n_{\lambda_1} & \mu^n_{\lambda_2} & \cdots & \mu^n_{\lambda_k} \end{bmatrix} \tag{2}$$

Here, $C_n$ represents the concentration of the nth component, $PA_k$ denotes the fluence-compensated PA image at the kth wavelength, and $\mu^n_{\lambda_k}$ is the normalized optical absorption coefficient of the nth component at the kth wavelength. Using these unmixed relative concentrations of HbO$_2$ ($m_{HbO_2}$) and Hb ($m_{HbO_2}$), the relative PA sO2 level mapping is calculated as

$$sO_2 = \frac{m_{HbO_2}}{m_{HbO_2} + m_{Hb}} \tag{3}$$

## Monte Carlo simulation of the optical fluence of the coplanar illumination scheme

A Monte Carlo numerical simulation of the optical fluence validated that the coplanar alignment of the optical and acoustic planes enabled by the US array transparency was superior to the conventional oblique alignment in terms of optical fluence. We used a MATLAB framework free software, MCmatlab[64], available at https://github.com/ankrh/MCmatlab. We simulated a 750 nm optical beam comprised of $10^7$ photons scattering into a $5 \times 5 \times 1.5$ cm$^3$ volume of biological tissue at 0.05 cm cubic resolution. The photons meeting the top and bottom boundary of the volume were set to escape. A x-y factorizable virtual optical beam was modeled following the actual physical parameters of an optical fiber- diffuser relay, including the 1 mm optical fiber diameter and 40° diverging angle of the engineered diffuser. The height

of the optical source was adjusted to fit within the lateral width of the TUT. While the coplanar illumination assumed the optical beam was at a 90° incidence angle to the tissue surface, for oblique illumination, we set the beam path at a 45° incidence angle and 4 mm distant from the US array. The 3D normalized fluence rate across the whole tissue block was calculated from the propagation results of all the released photons. The fluence-depth curve on the axial axis (x = 0 cm, y = 0 cm) was compared between the coplanar and oblique illumination.

### Indocyanine green dye injection

An ICG injection kit (TAIYO Pharma Tech Co., Ltd) was provided by Seoul National University Bundang Hospital. The kit included 25 mg of ICG powder and 10 mL of sterile water for injection. These were mixed to prepare a 2.5 mg/mL concentration of ICG dye solution. For the animal experiments, this solution was diluted to a concentration of 0.78 mg/mL (1 mM) with phosphate-buffered saline, and 0.1 mL was injected into the paw of the animals. In the human experiments, the original 2.5 mg/mL concentration of ICG dye solution was administered to the hand or foot, based on the physician's discretion.

### Animal and human experiments

The animal imaging experiments were conducted according to a laboratory animal protocol approved by the Institutional Animal Care and Use Committee of Pohang University of Science and Technology (POSTECH-2024-0025). Six-week-old healthy female Sprague-Dawley rats were used to visualize the lymphatic vessels and blood vessels near the axillary region, both before and after ICG injection. The laser fluence for all in vivo experiments was set at 10 mJ/cm², well below the American National Standards Institute safety threshold for skin exposure (20 mJ/cm²). The rats were initially anesthetized with 3% isoflurane via inhalation (1.0 L/min flow rate) and maintained on 1.5% isoflurane during imaging. A heating pad was used to keep the rat's body temperature at 36.5 °C, and US gel was applied to the ROI for acoustic coupling during USI/PAI/FLI.

For human experiments, all imaging procedures followed a protocol approved by the Seoul National University Bundang Hospital, Institutional Review Board (2024-02057). The experiment was conducted at Seoul National University Bundang Hospital, and we recruited four healthy male volunteers and two lymphedema patients (one female and one male). Before the experiment, all participants were provided with a comprehensive explanation of the study and signed to the informed consent statements, and the imaging experiments were performed on the patients by a plastic surgeon.

For both animal and human experiments, the DAQ and processing times are presented in Table S1 in the Supplementary Information.

### Reporting summary

Further information on research design is available in the Nature Portfolio Reporting Summary linked to this article.

## Data availability

All data that support the findings of this study are included in the manuscript and Supplementary Information. Any other relevant data are also available upon request from corresponding authors.

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

## Acknowledgements

This work was supported by the following sources: Basic Science Research Program through the National Research Foundation of Korea (NRF) funded by the Ministry of Education (2020R1A6A1A03047902, C.K.), NRF grant funded by the Ministry of Science and ICT (MSIT) (2023R1A2C3004880, C.K.; RS-2024-00339595, J.P.), Korea Health Technology R&D Project through the Korea Health Industry Development Institute (KHIDI), funded by the Ministry of Health & Welfare, Republic of Korea (RS-2024-00512879, C.K.), Korea Health Technology R&D Project through the Korea Health Industry Development Institute (KHIDI), funded by the Ministry of Health and Welfare (KH129481, Y.M.), Technology Innovation Program funded by the Ministry of Trade, Industry & Energy (MOTIE) (20024916, H.H.K.), and BK21 FOUR program (C.K.).

## Author contributions

J.P., D.O., J.Y., and C.K. conceived the idea, designed the study, and directed the project. J.P. and J.Y., H.H., and D.K. fabricated the TUT-array. H.H.K. contributed the TUT-array fabrication methods and tools. J.P., D.O., and C.K. designed the opto-US probe with TUT-array. J.P. and D.O. performed overall the experiments and analysed the data. Y.M. recruited healthy volunteers and patients for clinical experiments. J.P., D.O., and Y.M. performed clinical experiments. Y.M. and C.K. provided insightful advice. J.P., D.O., J.Y., and C.K. wrote the manuscripts.

## Competing interests

Chulhong Kim has financial interests in OPTICHO, which, however, did not support this work. Honghyeon Ha has financial interests in SonicLab, which did not support this work. All the others declare that they have no competing interests.
