## [Transparent Peer Review file · Nature Communications]

Clinical Ultrasound, Photoacoustic, and Fluorescence Image-guided Lymphovenous Anastomosis Microsurgery via a Transparent Ultrasound Transducer Array

Corresponding Author: Professor Chulhong Kim

Version 0:

Reviewer comments:

Reviewer #1

(Remarks to the Author)

The manuscript is well-written and presents a technically impressive and clinically meaningful study integrating photoacoustic, fluorescence, and ultrasound imaging via a transparent ultrasound transducer array. The methodology is clearly described, and the results demonstrate the strong potential of this triple-modal system in both preclinical and clinical settings.

However, a few critical points require clarification to strengthen the manuscript further:

1. Selection of Excitation Wavelengths:

The authors describe the use of different wavelength sets for in vitro (690, 720, 756, 796 nm) and in vivo (690, 756, 796, 820, 866 nm) imaging, yet the rationale for this distinction is not provided.

Please clarify why 720 nm was omitted and 820/866 nm were added for in vivo applications.

2. Imaging Time

The manuscript lacks information on the duration of each imaging procedure, particularly in clinical scenarios.

To better assess the practical applicability of this system, it would be helpful to include data on the average acquisition and processing times per patient or anatomical region.

Reviewer #2

(Remarks to the Author)

In this manuscript by Park et al., the authors present a 7 MHz transparent ultrasound linear array that combines ultrasound, photoacoustic, and fluorescence imaging into a single clinical probe – the opto-US probe. This successful integration signifies a significant technological and engineering advancement for transparent ultrasound transducers. The authors have properly and comprehensively characterized the opto-US probe and compared it with a commercial probe to demonstrate its sensitivity. Additionally, the demonstrated clinical application marks a major leap forward for clinical multimodal imaging and offers a wide range of potential clinical uses. Congratulations to the authors on achieving this powerful platform and its successful clinical applications. Please see my comments below for suggestions to further improve the manuscript:

1. The authors describe the advantages of expanding to 128 elements, which is indeed essential for a broader range of clinical applications. However, what are the limiting factors that prevented the authors from achieving a 128-element array? Is this due to fabrication complexity? If so, which step(s) pose the greatest challenge, and how might these be addressed or improved?

2. The authors re-poled the linear array after fabrication of the probe to enhance acoustic performance. Could the authors please include a detailed description of this process in the Methods section? Additionally, since re-poling is not a common step in clinically used commercial ultrasound probes, could the authors comment on why this step is necessary for the TUT-array? Furthermore, what is the laser pulse energy used during imaging, and did the authors observe any degradation in acoustic performance when the laser illuminated through the transparent ultrasound linear array?

3. The authors claim a 7 dB enhancement of SNR from the TUT-array (Fig. S7). However, this is not very clear from the

figure. When comparing the 10 mm target between the TUT-array and the commercial probe, it appears that the commercial probe may actually have a higher SNR.

4. Can authors comment on the feasibility to incorporate Doppler ultrasound imaging into this probe? Doppler imaging is one of the most commonly used ultrasound modalities, and its ability to provide deep vessel information could complement the shallower blood vessel data acquired by photoacoustic imaging (e.g., Fig. 5b, where UV and UA should be well observed using Doppler US).

5. Regarding Fig. S13, I appreciate the authors' demonstration of the advantage of using the TUT-array to achieve homogeneous illumination. However, could the authors explain why the black string phantom (Fig. S7) did not show such differences? Specifically, if angled illumination results in different light distribution, why does Fig. S7f show a similar level of SNR degradation from the TUT-array to the commercial probe across the full imaging depth?

6. The authors used TUT-array and opto-US probe interchangeably, please be consistent with the naming convention.

7. Many supplementary figures are missing color bars and/or scale bars. Please include these for clarity.

Version 1:

Reviewer comments:

Reviewer #1

(Remarks to the Author)
well revised.

Reviewer #2

(Remarks to the Author)
The authors addressed all my concerns very well. This is very impressive work. Congratulations again.

Title: Clinical Ultrasound, Photoacoustic, and Fluorescence Image-guided Lymphovenous Anastomosis Microsurgery via a Transparent Ultrasound Transducer Array

Authors

Jeongwoo Park^{1,3,†}
Donghyeon Oh^{1,†}
Jinhee Yoo^{1,†}
Honghyeon Ha^{1,5}
Donggyu Kim¹
Hyung Ham Kim^{1,*}
Yujin Myung^{2,*}
Chulhong Kim^{1,4,*}

Affiliations

¹Department of Electrical Engineering, Convergence IT Engineering, Mechanical Engineering, Medical Science and Engineering and Medical Device Innovation Center, Pohang University of Science and Technology (POSTECH), Pohang 37673, Republic of Korea

²Department of Plastic and Reconstructive Surgery, Seoul National University Bundang Hospital, Seoul National University College of Medicine, Seongnam 13620, Republic of Korea

³Department of Biomedical Convergence Science and Technology, Advanced Bioconvergence, and Cell and Matrix Research Institute, Kyungpook National University, Daegu, 41566, Republic of Korea

⁴Opticho Inc., Pohang, Republic of Korea

⁵SonicLab Inc., Siheung, Republic of Korea

†These authors have contributed equally

*Corresponding authors

Chulhong Kim
Email: chulhong@postech.edu

Yujin Myung
Email: surgene@snu.ac.kr

Hyung Ham Kim
Email: david.kim@postech.ac.kr

REVIEWER COMMENTS

Reviewer #1 (Remarks to the Author):

The manuscript is well-written and presents a technically impressive and clinically meaningful study integrating photoacoustic, fluorescence, and ultrasound imaging via a transparent ultrasound transducer array. The methodology is clearly described, and the results demonstrate the strong potential of this triple-modal system in both preclinical and clinical settings.

However, a few critical points require clarification to strengthen the manuscript further:

Comment R1-1: Selection of Excitation Wavelengths

The authors describe the use of different wavelength sets for in vitro (690, 720, 756, 796 nm) and in vivo (690, 756, 796, 820, 866 nm) imaging, yet the rationale for this distinction is not provided. Please clarify why 720 nm was omitted and 820/866 nm were added for in vivo applications.

***Reply:** Thank you for your closely observed comments. In the in vitro imaging experiments, our goal was to distinguish between blood and indocyanine green (ICG) signals in photoacoustic imaging. To this end, we selected four wavelengths (690, 720, 756, and 796 nm) that effectively capture the absorption spectra of the two chromophores. This spectral range was sufficient to achieve strong spectral separation without requiring an extended wavelength set. In contrast, the in vivo imaging of the human arm was designed to enable functional mapping of blood oxygen saturation (sO_2). For this purpose, the wavelength set was extended to include 820 nm and 866 nm in addition to 690, 756, and 796 nm. In this configuration, 796 nm was chosen as the isosbestic point of hemoglobin and serves as a neutral reference. Wavelengths below this point (690, 756 nm) and above this point (820, 866 nm) were chosen to provide sufficient spectral contrast to reliably separate oxygenated and deoxygenated hemoglobin. The 720 nm wavelength, which has an intermediate absorption between 690 and 756 nm, was intentionally excluded due to spectral redundancy. Instead, the inclusion of 820 and 866 nm, which are beyond the isosbestic point, increases spectral utilization for oxygenation status differentiation and improves optical penetration into deeper tissues. Since the total number of wavelengths directly affects imaging time, this configuration was chosen to optimize spectral resolution and acquisition efficiency.*

Based on this, we have updated the Methods section as follows.

“The latter configuration is optimized for sO_2 mapping by including 820 nm and 866 nm, which are located above the isosbestic point at 796 nm, and by excluding 720 nm due to its spectral redundancy between 690 and 756 nm.”

Comment R1-2: Imaging Time

The manuscript lacks information on the duration of each imaging procedure, particularly in clinical scenarios. To better assess the practical applicability of this system, it would be helpful to include data on the average acquisition and processing times per patient or anatomical region.

***Reply:** Thank you for the comment. We agree that including detailed time information is*

important for assessing practical applicability. As suggested, we have summarized the imaging parameters and the data acquisition and processing times for each experiment in Table S1 below.

We have updated the following sentences in the Methods section:

“For both animal and human experiments, the data acquisition and processing times are presented in Table S1 in the Supplementary Information.”

In addition, Table S1 has been added to the Supplementary Information.

Table S1. Imaging parameters, acquisition time, and post-processing time for each experimental condition.

	Animal	Human arm	Lymphedema patients
Imaging Modality	PA/US/FL	PA/US	PA/US/FL
Dimension	3D	3D	2D
Scanning strategy	Motor scanning	Motor scanning	Handheld
Field-of-view (mm)	45×30×40 mm ³ (Mosaic; X-, Y-, and Z-axis)	30×50×40 mm ³ (Mosaic; X-, Y-, and Z-axis)	20×40 mm ³ (X-, and Z-axis)
2D scan frame rate			
PA - single wavelength	20 Hz	20 Hz	20 Hz
Number of Wavelengths	4	5	5
PA – multispectral	5 Hz	4 Hz	4 Hz
US	20 Hz	20 Hz	20 Hz
Interleaved multispectral PA+US	5 Hz	4 Hz	5 Hz
FL camera	60 Hz (Live)	-	60 Hz (Live)
Number of frames per location	1	1	25
3D scan			
Elevational step	0.5 mm	0.5 mm	-
Number of Y-slices	60	100	1
Number of mosaic batches	4	2	-
Total acquisition time (acquisition, reconstruct & save)	8 min 48 sec	6 min 50 sec	30 sec

Reviewer #2 (Remarks to the Author):

In this manuscript by Park et al., the authors present a 7 MHz transparent ultrasound linear array that combines ultrasound, photoacoustic, and fluorescence imaging into a single clinical probe – the opto-US probe. This successful integration signifies a significant technological and engineering advancement for transparent ultrasound transducers. The authors have properly and comprehensively characterized the opto-US probe and compared it with a commercial probe to demonstrate its sensitivity. Additionally, the demonstrated clinical application marks a major leap forward for clinical multimodal imaging and offers a wide range of potential clinical uses. Congratulations to the authors on achieving this powerful platform and its successful clinical applications. Please see my comments below for suggestions to further improve the manuscript:

Comment R2-1: The authors describe the advantages of expanding to 128 elements, which is indeed essential for a broader range of clinical applications. However, what are the limiting factors that prevented the authors from achieving a 128-element array? Is this due to fabrication complexity? If so, which step(s) pose the greatest challenge, and how might these be addressed or improved?

***Reply:** Thank you for your insightful comment. The main limiting factor in scaling up to a 128-element array is the complexity of fabrication, particularly in the precise alignment and bonding of the optically transparent multilayer stack over a larger area. As the array size increases, maintaining uniform layer thickness and achieving consistent electrical interconnection across all elements become significantly more difficult. In addition, a larger bonding area makes it more challenging to ensure optical transparency and to maintain a highly polished surface finish across the entire multilayer structure. To address these technical challenges, precise control over such bonding parameters as pressure, adhesive distribution, and substrate flatness is required. In addition, high-resolution alignment systems and surface processing methods such as plasma activation or precision polishing may be necessary to maintain both optical transparency and a smooth interface across the expanded multilayer area.*

To reflect this point, we have updated the Discussion section as shown below:

“However, scaling up poses significant fabrication challenges, particularly in aligning and bonding the transparent multilayer stack over a larger area. As the array size increases, it becomes more difficult to maintain uniform thickness, ensure consistent electrical connections, and preserve optical transparency with a smooth surface finish. To address these issues, precise control of such bonding parameters as pressure, adhesive distribution, and substrate flatness is required. In addition to high-resolution alignment, surface treatment methods such as plasma activation or precision polishing may also be required.”

Comment R2-2: The authors re-poled the linear array after fabrication of the probe to enhance acoustic performance. Could the authors please include a detailed description of this process in the Methods section? Additionally, since re-poling is not a common step in clinically used commercial ultrasound probes, could the authors comment on why this step is necessary for the TUT-array? Furthermore, what is the laser pulse energy used during imaging, and did the authors observe any degradation in acoustic performance when the laser illuminated through the transparent ultrasound linear array?

Reply: *Thank you for the valuable comments. While the re-poling process is not typically required for commercial ultrasound probes, it becomes increasingly necessary for probes fabricated using single-crystal piezoelectric materials such as PMN-PT. Compared to conventional piezoelectric ceramics like PZT, PMN-PT exhibits a lower depoling temperature (~60 °C to 95 °C), a lower Curie temperature (~130 °C to 178 °C) [Ref 1], and greater mechanical fragility [Ref 2]. These characteristics make PMN-PT more susceptible to depolarization during fabrication processes, especially during bonding and dicing. In our case, fabricating the TUT-array involves complexity beyond that of standard commercial probes due to the requirement for optical transparency. This includes multilayer stacking, precision bonding under controlled pressure, and fine-pitch dicing, all of which can induce thermal and mechanical stress. Such conditions increase the likelihood of partial or complete depolarization of the piezoelectric elements. Therefore, the re-poling step is essential to recover optimal piezoelectric performance. The effectiveness of this step is supported by improved dielectric behavior, as confirmed by the capacitance measurements shown in Figure S1 of the Supplementary Information.*

[Ref 1] Li, X. et al. Micromachined PIN-PMN-PT crystal composite transducer for high-frequency intravascular ultrasound (IVUS) imaging. *IEEE transactions on ultrasonics, ferroelectrics, and frequency control* 61, 1171-1178 (2014).

[Ref 2] Kim, H.-P. et al. Electrical de-poling and re-poling of relaxor-PbTiO₃ piezoelectric single crystals without heat treatment. *Nature communications* 15, 6420 (2024).

As stated in the manuscript, the laser fluence used during all in vivo imaging experiments was set to 10 mJ/cm², which is well below the ANSI safety limit for skin exposure (20 mJ/cm²).

Regarding potential degradation in acoustic performance during laser illumination, we did not observe any noticeable deterioration in the performance of the TUT-array. This stability is attributed to two main factors: (1) the low laser pulse energy used, which minimizes thermal stress, and (2) the high optical transmittance of the TUT-array components, which allows the laser to pass through with minimal absorption or scattering. Specifically, the TUT-array exhibited an optical transmittance of approximately 72.7% at a wavelength of 720 nm, as shown in Supplementary Figure S3. This high transparency ensured that the laser energy was efficiently transmitted to the target tissue without significant deposition within the array itself. Additionally, the laser light was diffusely delivered, further mitigating the risk of localized heating or mechanical disruption. As a result, ultrasound propagation and imaging performance remained stable throughout the experiment.

To reflect your suggestion, we have provided a detailed description of the re-poling process in the Methods section of the revised manuscript:

“Re-poling process

Re-poling was conducted to restore the polarization of the elements, which may have been degraded by thermal and mechanical stresses during bonding and dicing. Re-poling was performed after the TUT-array was completely assembled, including electrical connections via FPCBs and external cables. This step was particularly necessary due to the use of single-crystal materials and the complex bonding procedures required for optical transparency. A DC high-voltage supply (Kistech, Republic of Korea) was connected to each element through the signal and ground lines of the cable. To maintain the original polarization direction, a positive bias was applied to the ground line and a negative bias to the signal line. A high voltage (~180 V) was applied for 60 seconds per element under ambient conditions in air. Capacitance was measured before and after re-poling to assess changes in dielectric properties and confirm electrical recovery.”

Comment R2-3: The authors claim a 7 dB enhancement of SNR from the TUT-array (Fig. S7). However, this is not very clear from the figure. When comparing the 10 mm target between the TUT-array and the commercial probe, it appears that the commercial probe may actually have a higher SNR.

Reply: *Thank you for your perceptive comment. We understand and agree with your observation that the SNR difference in Figure S7 may not be readily apparent. To address this, we revisited the experiments and normalized the images to the background noise level instead of the peak signal. From these updated images, now provided in Fig. S7c, the SNR difference is clearly visible at 10 mm depth, with the TUT-array showing an improvement of approximately 8 dB over the commercial probe with oblique illumination. Furthermore, the average SNR improvement across all depths is about 7 dB, as depicted in Fig. S7f. Given the wide 20–70 dB dynamic range of the images, we changed the colormap from ‘hot’ to ‘jet’ to enhance visual contrast and better highlight the pixel intensity differences.*

*Regarding the SNR comparison, the improved SNR is an integrated result of the adequate acoustic sensitivity of the UT array and its superior optical delivery yield, as is extensively described in the reply to **Comment R2-5** below.*

We have revised the Supplementary Information to include these updated figures for clarity.

Fig. S7. Performance evaluation of photoacoustic imaging (PAI) by the opto-US probe. (a) Cross-section of black strings in a tissue-mimicking phantom. (b) Schematic comparison of the TUT-array with an optical fiber in coaxial alignment and a commercial opaque UT with an optical fiber in oblique alignment. (c) Comparison of PA B-scan images between the TUT-array with coaxial alignment and the commercial UT with oblique alignment. Comparison of the (d) axial and (e) lateral PAI resolutions, and (f) PA SNR between the TUT-array and the commercial UT.

Comment R2-4: Can authors comment on the feasibility to incorporate Doppler ultrasound imaging into this probe? Doppler imaging is one of the most commonly used ultrasound modalities, and its ability to provide deep vessel information could complement the shallower blood vessel data acquired by photoacoustic imaging (e.g., Fig. 5b, where UV and UA should be well observed using Doppler US).

Reply: *We thank the Reviewer for this thoughtful comment. We agree that Doppler ultrasound imaging could complement photoacoustic imaging by providing deeper vascular information, contributing to a more comprehensive assessment of tissue hemodynamics. Although Doppler US was not implemented in the present study, its integration into our current system is conceptually feasible. Real-time Doppler imaging requires a longer acquisition period for repeated high-frame-rate sampling, as well as greater computational resources for real-time image reconstruction and clutter filtering of consecutive frames. In our current framework, we prioritized real-time performance for multispectral photoacoustic/ultrasound imaging, which imposes strict timing constraints, particularly the 50 ms interval between laser pulses at 20 Hz. Given these limitations, incorporating a full Doppler acquisition sequence within this time window remains technically challenging. However, future advances in GPU-based processing and acquisition optimization may enable such integration. Combining Doppler US for blood flow assessment with PA-based spectral contrast mapping could ultimately allow for comprehensive metabolic rate monitoring.*

We view this as a promising direction for future development and have included this outlook in the revised Discussion section as below:

“Acknowledging the complementary strengths of PA and US imaging, the implementation of advanced functional techniques holds strong potential to enhance the diagnostic utility of this dual modality system, as demonstrated by multispectral PA imaging in this study. For instance, integrating Doppler US imaging could enable simultaneous assessment of blood flow and oxygenation, providing a pathway toward real time comprehensive metabolic rate imaging. Future work should focus on addressing technical challenges, particularly the need to accommodate the full Doppler US sequence including successive acquisitions, image reconstruction, and clutter filtering within the laser pulse intervals. These challenges may be addressed through tailored optimization of acquisition parameters and GPU based parallel computing.”

Comment R2-5: Regarding Fig. S13, I appreciate the authors' demonstration of the advantage of using the TUT-array to achieve homogeneous illumination. However, could the authors explain why the black string phantom (Fig. S7) did not show such differences? Specifically, if angled illumination results in different light distribution, why does Fig. S7f show a similar level of SNR degradation from the TUT-array to the commercial probe across the full imaging depth?

Reply: *We appreciate the opportunity to clarify the comparisons presented in our study. Although both Figures S7 and S13 compare images acquired under coaxial and oblique illumination, please note that they have different implications.*

Figure S13 highlights the advantage of coaxial illumination in 3D imaging of irregular surfaces, where oblique beams can be obstructed by variations in surface height and slope, resulting in inhomogeneous PA signal intensity across the scanned volume. This consideration becomes an exclusive advantage for the TUT-array, facilitating robust 3D tomographic imaging across various complex geometries.

On the other hand, **Figure S7** presents a comparison between our custom-fabricated TUT-array and a commercial opaque UT array. Although they have physical form factors, there is a fundamental acoustic sensitivity difference between the custom-made TUT array and the commercial opaque UT array, due to differences in material and fabrication processes adjusted for optical transparency. As presented in **Figure S6**, the commercial array demonstrated higher acoustic sensitivity than the TUT array. Conversely, light can be more efficiently delivered with coaxial illumination through the TUT array by minimizing the optical path length, in contrast to the oblique illumination which remains the only feasible option for opaque UT arrays. The Monte Carlo simulations in **Figure S12** further validate the improved optical fluence from coaxial delivery rather than oblique delivery. In consequence, as demonstrated in Fig. S7, the TUT array achieves comparable, but greater, SNR in PA imaging, because its lower sensitivity is compensated for by more efficient light delivery.

To address the discrepancy between SNR degradation and simulated light distribution, a fairer comparison would involve using the same TUT-array to eliminate the influence of acoustic sensitivity, while varying only the illumination configuration. To this end, we imaged a 0.1-mm black nylon string pattern submerged in a turbid 1% TiO₂ aqueous solution using the identical TUT-array, once with coaxial illumination and once with oblique illumination. Compared to oblique illumination, the coaxial configuration yielded a higher PA amplitude and an SNR improvement of 8 dB,. Additionally, the coaxial image exhibited a gradual signal decrease with depth, consistent with expected optical attenuation, whereas the oblique illumination resulted in lower signal intensity even in the near field, likely due to non-uniform light delivery caused by surface scattering.

To reflect this additional validation, we have revised the manuscript by including Supplementary Figure S8 and the following description:

“While **Figure S7** shows comparable SNR degradation between the TUT-array and the commercial probe, this result reflects both the acoustic sensitivity and the light delivery efficiency. To clarify the specific impact of coaxial versus oblique illumination, we performed an additional phantom experiment using the same TUT-array under both configurations, demonstrating that coaxial illumination yields a higher SNR of 8 dB and more uniform signal decay with depth (**Fig. S8, Supplementary information**).”

Fig. S8. Comparison of US and PA B-scan images acquired using the TUT-array with coaxial versus oblique laser illumination.

Comment R2-6: The authors used TUT-array and opto-US probe interchangeably, please be consistent with the naming convention.

Reply: *We appreciate your attention to terminology. In our manuscript, the term TUT-array refers specifically to the transparent ultrasound transducer component itself, while opto-US probe denotes the complete handheld assembly incorporating the TUT-array along with optical and acoustic components. We have clarified this distinction in the manuscript and carefully reviewed all instances to ensure consistent usage throughout.*

Comment R2-7: Many supplementary figures are missing color bars and/or scale bars. Please include these for clarity.

Reply: *We apologize for the missing parts. We have added color bars and scale bars to the relevant supplementary figures to provide more clarity and complete information.*